

# Basic psychological need satisfaction and aggressive behavior: the role of negative affect and its gender difference

Fen Dou[1], Qinglin Wang[2], Minghui Wang[2], Entao Zhang[2] and Guoxiang Zhao[2]

[1] School of Psychology, Nanjing Normal University, Nanjing, China
[2] School of Psychology, Henan University, Kaifeng, China

## ABSTRACT

**Background**. Basic psychological need satisfaction (BPNS) is a significant factor in a person's development, especially for adolescents, and the failure to satisfy these basic needs may contribute to individuals' aggressive behavior. However, it is still unclear about the underlying mechanism by which BPNS is negatively associated with aggressive behavior. This study aimed to explore the relationship between BPNS and aggressive behavior in Chinese adolescents, with a focus on the mediating role of negative affect and its gender differences.

**Method**. A sample of 1,064 junior high school students from three schools in China were selected randomly for the cross-sectional survey. The revised Need Satisfaction Scale, the Positive and Negative Affect Schedule, and Youth's Self-Report were used to measure BPNS, affect, and aggressive behavior. The proposed model was examined by the structural equation modeling test and multi-group comparison analysis.

**Results**. The results showed that BPNS was negatively linked with adolescents' aggressive behavior, and this effect was mediated by negative affect. Moreover, multigroup analysis demonstrated that there existed a stronger negative association between BPNS and negative affect in female group. Also, the mediating effect of negative affect in the model was greater for girls.

**Conclusions**. Our findings highlighted the importance of BPNS in adolescents' social behavior (*i.e.*, aggressive behavior), and reveal disparate patterns in how BPNS affects aggressive behavior in girls as compared to boys.

## INTRODUCTION

Aggressive behavior refers to any act or behavior that causes physical or psychological harm to others in a direct or indirect way (*Anderson & Bushman, 2002*). Aggressive behavior may have significant adverse effects on adolescents' health (*Velotti et al., 2017*), shaping their social interactions and social adjustment into adulthood (*Ettekal & Ladd, 2017*). As youth enter adolescence, the rate of aggressive behavior seems to be sharply increasing (*Chung et al., 2019*), and the increasing prevalence of aggressive behavior among adolescents has become a major social and health concern globally (*Valois, Zullig & Revels, 2017*). Due to the influence of Collectivism and Confucianism, Chinese culture places special emphasis

Corresponding author
Minghui Wang,
wmhwang@henu.edu.cn

on maintaining group harmony (*Li, Xie & Shi, 2012*), whereas aggressive behavior regarded as "highly problematic and abnormal" is culturally inhibited and might be more likely not be accepted by others (*Yang et al., 2022*). However, the risk and protective factors of aggressive behavior in Chinese adolescence are still poorly understood. Thus, it is urgent to pay more attention to Chinese adolescents' aggressive behaviors, and it is necessary to identify the influencing factors and explore potential mechanisms that lead to adolescents' aggressive behavior, which can provide an empirical basis and theoretical guidance for reducing aggressive behavior in adolescents.

Self-determination theory (SDT) holds that there are three basic psychological needs for humans: the need for autonomy, competence, and relatedness (*Deci & Ryan, 2000*). Basic psychological need satisfaction (BPNS) refers to the satisfaction or fulfillment of individuals' basic psychological needs (*i.e.,* need for autonomy, competence, and relatedness), which are defined as the integration between innate psychological needs and support from the social environment (*Ryan & Deci, 2004*). In addition, BPNS is conceptualized as the provision of innate psychological "nourishments" to preserve ongoing psychological growth and well-being (*Ryan & Deci, 2000*). BPNS is essential to a person's development, especially for adolescents (*Chen et al., 2015*). When individuals are brought up in a positive environment which satisfy their basic psychological needs, this stimulates their intrinsic motivation to seek positive development, thus promoting their mental health and social development (*Deci & Ryan, 2008*). Conversely, if one's basic psychological needs remain unmet, this may lead to a psychological imbalance that cultivates hostile and antisocial behaviors (*e.g.*, aggressive behavior) (*Yu, Li & Zhang, 2015*). A plausible theoretical explanation for negative effect of BPNS on aggressive behavior is provided by the conservation of resources theory (COR, *Hobfoll et al., 2018*). The fourth principle of COR theory states that individuals enter a defensive mode to preserve the self that is often aggressive and may become irrational when their resources are outstretched or exhausted (*Hobfoll et al., 2018*). According to COR theory, BPNS, as one of personal resources, can be a distal determinant of individuals' aggressive behavior. Empirically, extant studies have found a negative association between BPNS and antisocial behavior (*e.g.*, aggressive behavior) in adolescents (*Sun et al., 2021*), that is, impede individuals' BPNS may lead to aggressive behavior (*Choe & Read, 2019*). This provides the basis for our contention that BPNS can be a negative predictor of aggressive behavior in adolescents.

Simply focusing on BPNS as the sole determinant of aggressive behavior in adolescents is inadequate (*Zhou, Ntoumanis & Thogersen-Ntoumani, 2019*). The mediating mechanisms through which BPNS impact aggressive behavior remains unclear. The exploration of this question will help us better understand how BPNS impacts aggressive behavior.

In previous studies, positive and negative affect have emerged as two independent dimensions (*Watson, Clark & Tellegen, 1988*). Positive affect (PA) reflects the extent to which a person feels enthusiastic, active, and alert, in contrast, negative affect (NA) is a general dimension of subjective distress and unpleasurable engagement that subsumes a variety of aversive mood states (*Watson, Clark & Tellegen, 1988*). According to COR theory (*Hobfoll et al., 2018*) and the general aggression model (GAM, *Anderson & Bushman, 2002*), when resources are scarce (*i.e.*, adverse situational factor), individuals will experience more

negative affect. Served as one type of personal resource, BPNS is negatively linked with negative affect (*Gui, Kono & Walker, 2019*). When basic needs are not satisfied, individuals would struggle to cope with challenges and exhibit negative affect (*Auclair-Pilote et al., 2021*).

Furthermore, GAM proposes that external negative events can cause negative affect, which may further lead to aggressive behavior (*Anderson & Bushman, 2002*), and this is consistent with the viewpoint of the cognitive-neoassociationistic model (*Berkowitz, 1993*), suggesting that external stress situations indirectly influence aggressive behavior through the role of internal state (*e.g.*, affect; *Anderson & Bushman, 2002*). Meanwhile, previous literature provides empirical support that negative affect is often associated with more aggressive behavior (*e.g.*, *Zhu et al., 2020*). When individuals' resources were insufficient, negative affect may increase, and further lead to aggressive behavior (*Kelber, Lickel & Denson, 2020*). Accordingly, it is reasonable to hypothesize that negative affect would play a mediating role between BPNS and adolescents' aggressive behavior. Moreover, based on COR theory, individuals who have most resources or have experienced the least resource loss are more likely to show positive affect (*Doane, Schumm & Hobfoll, 2012*). Thus, from the perspective of COR theory, BPNS can be considered a personal resource that is positively associated with positive affect. Some empirical studies have shown that BPNS was associated with more positive affect (*e.g.*, *Schutte & Malouff, 2021*; *Tang, Wang & Guerrien, 2020*). However, limited empirical studies showed that compared with negative affect, positive affect has weaker or no effect on aggressive behavior (*e.g.*, *Chester, 2017*). Thus, we assume that positive affect would not mediate between BPNS and adolescents' aggressive behavior when both positive affect and negative affect are considered in one model.

According to GAM (*Anderson & Bushman, 2002*) and social role theory (*Doherty & Eagly, 1989*), gender serves as an individual factor (*i.e.,* a biological variable) that may interact with the situation in determining aggressive behavior, and individuals of different sexes have different gender role expectations. Gender role expectations can provide us the theoretical framework for understanding the potential gender differences in the relationship between social behaviors and other variables (*Orue, Calvete & Gamez-Guadix, 2016*). However, to our best knowledge, less is investigated in terms of the possibility that the BPNS–affect–aggressive behavior link may differ depending on adolescents' gender. It would be valuable to explore this striking gap among adolescents. In fact, there exist gender differences between males and females in emotional experience due to various factors, such as living habits and gender cognition (*Hu et al., 2023*). For instance, extant studies found that women are more emotionally sensitive, and more prone to emotional problems (*Zhao et al., 2020*). *Shao et al. (2018)* found that BPNS can serve as a stronger negative predictor of unfavorable emotional outcomes for girls relative to that observed for boys. Thus, different patterns in the relationship between BPNS and negative affect may exist in males compared to females.

Furthermore, a large body of research have explored the gender role in aggressive behavior. However, few studies have compared the effects of BPNS and affect on aggressive behavior between males and females directly, and the findings about gender effects on

aggressive behavior are still inconclusive. Some research found males are more often associated with various types of overt aggressive behavior and females were more likely to display indirect aggression (*e.g.*, social exclusion *Borau & Bonnefon, 2019*), while others revealed reversed findings (*e.g.*, girls were more susceptible to direct aggression, *Zhang, Liu & Zhang, 2020*) or no gender difference on general aggressive behaviors (*Kang et al., 2021*). One possible explanation for the inconsistent findings is that existing studies has explored the relationships between different variables and aggressive behavior in different samples. Despite these inconsistencies in the domain of aggression, we mainly focused on Chinese adolescents' general aggressive behavior in this study, and try to explore the potential gender differences in the relation between BPNS and aggressive behavior or the mediating effect of affect. Given that females are more sensitive to external information, more likely to use emotion-focused coping strategies (*Armour et al., 2011*), and show more emotional problems than males (*Zhen, Yao & Zhou, 2022*), female adolescents may show more aggressive behavior when we are taking negative emotional factors into consideration. That is, compared with males, females might be more susceptible to the low levels of BPNS, and vulnerable to the negative affect, which may further lead to more aggressive behavior.

Thus, based on GAM, social role theory and extant empirical evidence about gender differences on negative affect and aggressive behavior, we assume that gender may moderate the relationship between BPNS and adolescents' negative affect, which further could exert an effect on their aggressive behavior. That is, it is reasonable to assume that some gender differences may be observed in the mediating effect of negative affect. More specifically, the effect of BPNS on negative affect would be greater in female adolescents than that in male adolescents, and similarly the indirect effect of BPNS on aggressive behavior through negative effect would be greater for girls than that for boys.

## The present study

Based on the theoretical framework of the conservation of resources theory (COR) and the general aggression model (GAM), the present study constructs a moderated mediation model to examine whether negative affect mediates the relationship between BPNS and aggressive behavior, and whether there exist gender differences in the mediating effect in a sample of Chinese adolescents (*i.e.*, junior students from three public junior middle schools). It is worth mentioning that positive affect is also examined in our proposed model, even though we focus mainly on negative affect in current study. This study can provide insights into the relationship between Chinese adolescents' BPNS and aggressive behavior, and an investigation of the potential mechanism might be conducive to fully grasping the nature of the association. Base on the theoretical frameworks and existing empirical work mentioned above, the current study proposes three research hypotheses:

H1: BPNS is negatively associated with Chinese adolescents' aggressive behavior.

H2: Negative affect plays a mediating role between BPNS and aggressive behavior in Chinese adolescents.

H3: Gender moderates the association between BPNS and negative affect, and further plays a moderating role in the mediating mechanism of negative affect. Compare with male adolescents, both the effect of BPNS on negative affect and the indirect of BPNS on

aggressive behavior are greater for female adolescents. In addition, regarding the potential gender differences in other pathways of the proposed mediation model, we did not propose specific hypothetical patterns due to the lack of sufficient empirical research, but instead conducted an exploratory analysis.

## MATERIALS & METHODS

### Participants and procedure

We recruited 1,100 students from three public junior middle schools in Nanjing, a southeast city in China. Data were collected in the classroom with trained psychological graduate students administering the paper-and-pencil questionnaires. All participants attended the study voluntarily and all the participants' parents gave oral informed consent before the survey. The survey emphasized that responses were anonymous and confidential. It took approximately ten minutes to complete all items. The study protocol was approved by the Psychology Research and Ethics Committee at Henan University in China (ID: 2020914).

Finally, we obtained a total of 1064 valid questionnaires, with a response rate of 96.73%. The participants were aged from 13 to 16 years ($M$ age $= 14.25$ years, $SD = 0.96$). Among them, 454 were males, and 610 were females. There were 37.50% seventh-grade students, 32.42% eighth-grade students and 30.08% ninth-grade students. The authors' university ethics committee approved the current research.

### Measures

#### Basic psychological need satisfaction

The Revised Need Satisfaction Scale, developed by *La Guardia et al. (2000)*, was used to assess the BPNS. This scale was validated in college students by *La Guardia et al. (2000)*, and showed adequate reliability and validity. The authors have permission to use this instrument from the copyright holders. There were nine items in this scale which involved three dimensions of the need for autonomy (*e.g.*, "I feel free to be who I am."), competence (*e.g.*, I feel very capable and effective.") and relatedness (*e.g.*, "I feel loved and cared about."). Three of the nine items are reverse scored and a 7-point Likert scale (1 = strongly disagree, 7 = strongly agree) was used. The scale items were translated into Chinese and back-translated to facilitate respondents' understanding. In this study, internal reliability for the scale was good ($\alpha = 0.87$). The confirmatory factor analysis (CFA) showed $\chi^2/df = 3.24$, RMSEA $= 0.066$, CFI $= 0.97$, TFI $= 0.94$, SRMR $= 0.028$, indicating satisfactory reliability and validity of the scale.

#### Positive affect and negative affect

The Chinese Positive and Negative Affect Schedule (PANAS, *Huang, Yang & Ji, 2003*) were used to measure positive affect and negative affect. This scale consists of 20 adjectives involving two dimensions of positive affect (PA; *e.g.*, "excited", "inspired" and "active") and negative affect (NA; *e.g.*, "distressed", "nervous" and "hostile"). Each dimension contained 10 adjectives and ranked using a seven-point Likert scale (1 = not at all true of me, 7 = exactly true of me). The original PANAS was developed and validated by *Watson, Clark & Tellegen (1988)*, showing satisfactory reliability and validity. In current study, the

internal consistency coefficients were 0.93 and 0.91 for the positive and negative affect subscale, respectively. The results of confirmatory factor analysis (CFA) were satisfactory with $\chi^2/df = 2.79$, RMSEA $= 0.058$, CFI $= 0.93$, TFI $= 0.92$, SRMR $= 0.047$.

### Aggressive behavior

The Chinese version of the Aggressive Behavior subscale (*Leung et al., 2006*) from the Youth's Self-Report (YSR, *Achenbach & Dumenci, 2001*) was used in this study to measure participants' aggressive behavior. The aggressive subscale in YSR consists of 19 items, and an example item is "I destroy others' belongings." Participants rated the frequency of each statement for them during the last six months on a seven-point Likert scale (1 = almost never, 7 = almost always). The original subscale has been validated in Chinese adolescents and showed adequate reliability and validity (*Leung et al., 2006*). The authors have permission to use this instrument from the copyright holders. In the current study, the internal reliability for the scale was satisfactory ($\alpha = 0.91$), with good CFA fit indices: $\chi^2/df = 1.96$, RMSEA $= 0.062$, CFI $= 0.96$, TFI $= 0.93$, SRMR $= 0.058$.

### Data analysis

The collected data were checked and the valid data entered into SPSS 21.0 software (SPSS Inc., Chicago, IL, USA) for analysis. The preliminary analyses were conducted by SPSS software, and the validity test was carried out by confirmatory factor analysis (CFA) with MPLUS 7.0. In addition, AMOS 26.0 was used for testing the proposed structural equation model (SEM) and multi-group comparison analysis.

## RESULTS

### Common method bias test

We conducted the Harman single factor test to identify whether severe common method bias (CMB) was present or not (*Podsakoff et al., 2003*). Results showed that the single factor model fit was very poor ($\chi^2/df = 20.83$, RMSEA $= 0.096$, CFI $= 0.47$, TLI $= 0.45$, SRMR $= 0.124$), indicating that CMB is not a serious issue in our sample data.

### Preliminary analyses

The descriptive statistics and correlations among variables were shown in Table 1. The results showed that BPNS negatively related to negative affect and aggressive behavior, and positively related to positive affect. Meanwhile, negative affect positively related to aggressive behavior, whereas positive affect negatively related to aggressive behavior and negative affect.

Additionally, age and gender emanated various degrees of correlations with the four latent variables. Also, the results of one-way ANOVA showed significant grade differences for BPNS ($F$ (2, 1061) $= 19.872$, $p < 0.001$), aggressive behavior ($F$ (2, 1061) $= 11.473$, $p < 0.001$), negative affect ($F$ (2, 1061) $= 6.727$, $p < 0.01$), and positive affect ($F$ (2, 1061) $= 11.698$, $p < 0.01$). As a result of these preliminary findings, gender, age, and grade were controlled in subsequent analyses.
**Table 1 The descriptive statistics and correlations among variables.**

| Variables | M | SD | 1 | 2 | 3 | 4 | 5 |
|---|---|---|---|---|---|---|---|
| 1. BPNS | 4.92 | 1.35 | 1 | | | | |
| 2. Positive affect | 4.94 | 1.52 | 0.61*** | 1 | | | |
| 3. Negative affect | 2.48 | 1.36 | −0.49*** | −0.42*** | 1 | | |
| 4. Aggressive behavior | 2.01 | 0.88 | −0.34*** | −0.26*** | 0.57*** | 1 | |
| 5. Age | 14.25 | 0.96 | −0.11** | −0.06* | 0.06 | 0.08** | 1 |
| 6. Gender | 1.57 | 0.50 | −0.12** | −0.23*** | 0.15*** | 0.07* | 0.01 |

**Notes.**

Male was coded as 1, female was coded as 2.

*$p < 0.05$.

**$p < 0.01$.

***$p < 0.001$.

## Testing the mediating role of negative affect

Structural equation modeling (SEM) was used to test the mediating effect of affect between BPNS and aggressive behavior in adolescents. BPNS served as an exogenous latent variable while positive affect, negative affect and aggressive behavior were constructed as endogenous latent variables. We use the item parceling method to address the issue of the latent variables containing too many observation indicators (*Landis, Beal & Tesluk, 2000*), which can simplify the model and improve the goodness of fit of the model (*Matsunaga, 2008*). Following earlier research (*Matsunaga, 2008*), three parcels per factor can provide more stable parameter estimates and are preferred. The CFA results showed that original three-factor structure of the BPNS scale had good fit in current study (see 'Measures'). Thus, for BPNS, internal-consistency approach was used and the three dimensions were packaged separately, resulting in three parcels (*i.e.*, N1, N2, and N3 showed in Fig. 1) for BPNS. Since the measures of positive affect, negative affect and aggressive behavior are all unidimensional, we created three parcels for each latent construct (*i.e.*, three parcels of A1, A2, A3 for negative affect; three parcels of A4, A5, A6 for positive affect; three parcels of Y1, Y2 and Y3 for aggressive behavior showed in Fig. 1) by the random algorithm as recommended by *Matsunaga (2008)*.

The results showed that the direct path coefficient from BPNS to aggressive behavior was significant ($\beta = -0.37$, $t = -10.63$, $p < 0.001$) in the absence of mediators. After adding potential mediating variables (*i.e.*, positive affect and negative affect) into the model, the mediated model (see Fig. 1) revealed a good fit to the data ($\chi^2/df = 4.59$, CFI = 0.97, TLI = 0.96, RMSEA = 0.058). Results showed that BPNS were positively and negatively associated with positive affect ($\beta = 0.68$, $t = 20.89$, $p < 0.001$) and negative affect ($\beta = -0.56$, $t = -16.38$, $p < 0.001$), respectively. Meanwhile, negative affect was positively associated with aggressive behavior ($\beta = 0.63$, $t = 16.08$, $p < 0.001$), but the relationship between positive affect and aggressive behavior was not significant ($\beta = 0.05$, $t = 1.18$, $p = 0.24$). In addition, the path coefficients from BPNS to aggressive behavior became non-significant ($\beta = -0.05$, $t = -1.14$, $p = 0.25$). This suggests that negative affect plays a full mediating role between BPNS and aggressive behavior among adolescents.

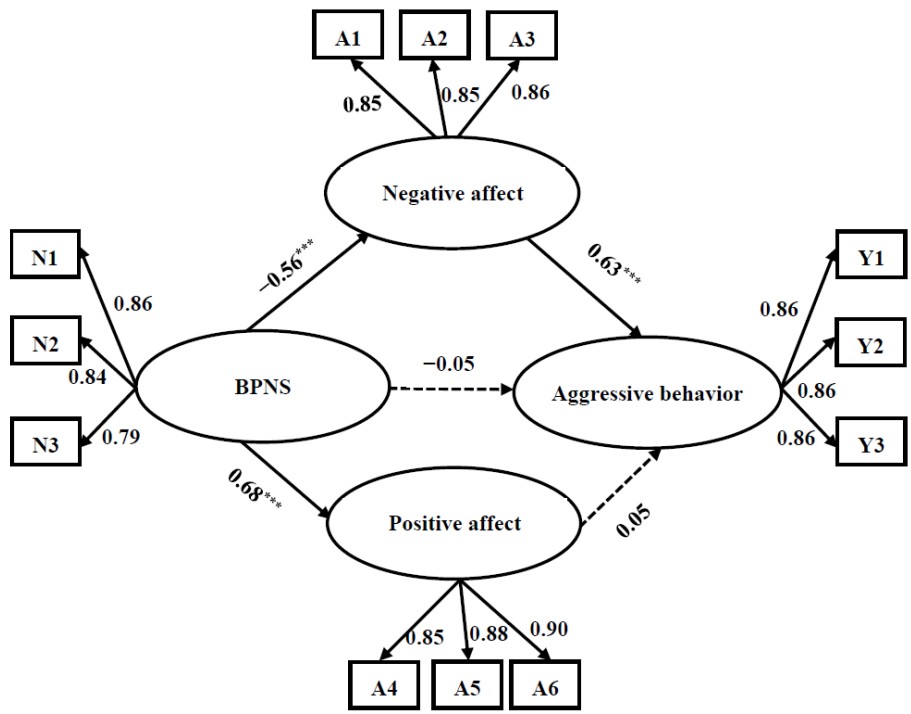

**Figure 1** **The mediating effect of positive affect and negative affect between BPNS and aggressive behavior.** Factor loading and path coefficients are standardized. N1–N3 = three parcels of BPNS; A1–A3 = three parcels of negative affect; A4–A6 = three parcels of positive affect; Y1-Y3 = three parcels of aggressive behavior. Control variables were not presented for the sake of clarity. *$p < .05$, **$p < .01$, ***$p < .001$.

A bias-corrected bootstrap test (a bootstrap sample of 2,000) was conducted to further examine the significance of the mediating effect. Results indicated that the mediating effect of positive affect was not significant, with the 95% confidence interval (CI) of the path "BPNS → positive affect → aggressive behavior" (−0.07, 0.13) containing zero. While the negative affect was a significant mediator in the structural equation model, the 95% CI of "BPNS → negative affect → aggressive behavior" path (−0.40, −0.24) did not contain zero. The value of the mediating effect was $−0.56 \times 0.63 = −0.35$, Boot $SE = 0.04$.

## Multiple-group comparison analysis

A multi-group analysis was used to identify whether the path coefficients significantly differed between male and female students. SEM was used to establish models for both male and female groups respectively. The model fit indices in our male sample were $\chi^2/df = 2.40$, CFI = 0.97, TLI = 0.96, RMSEA = 0.061, and in the female sample, the fit indices were $\chi^2/df = 2.429$, CFI = 0.98, TLI = 0.97, RMSEA = 0.055. In general, the model fits in both samples were satisfactory, meeting the condition of multi-group analysis (*Hou, Wen & Cheng, 2004*).

Then, multigroup structural equation modeling was used to set equivalence models. An unrestricted model (M1) was initially set for the multiple-group comparison, and then freely estimated two gender groups. Following this, a structural weights model (M2) was

**Table 2  Multiple-group comparison analysis of mediation model.**

| Model | $\chi^2$ | df | RMSEA | CFI | TLI | NFI |
|---|---|---|---|---|---|---|
| M1 | 440.470 | 142 | .044 | .967 | .957 | .952 |
| M2 | 473.274 | 156 | .044 | .965 | .959 | .948 |

Notes.
   M1: Unconstrained model, M2: Structural weights model.

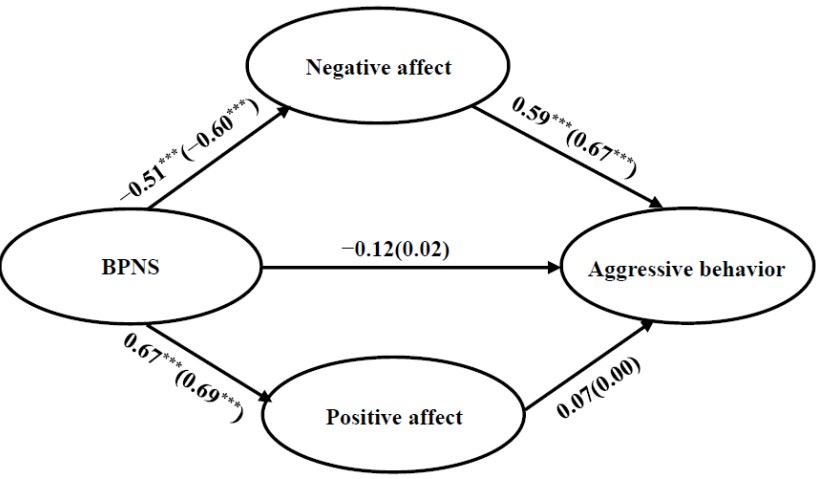

**Figure 2  The result of multiple-group comparison analysis which was produced to examine the gender differences.** Path coefficients are standardized, females are inside the parentheses, outside the parentheses are males.

constructed to examine the gender differences. In this model, we constrained the structural paths across the sexes to be equal. The fit indices of the two models were satisfactory (see Table 2). Compared with M1, M2 has $\Delta\chi^2 = 32.80$, $\Delta df = 14$, $p < 0.01$, which indicates that there exists a significant gender difference in the mediating model (see Fig. 2).

To further test the specific gender differences, the critical ratios of differences (CR) were calculated (*Arbuckle, 2003*). The path coefficient "BPNS → negative affect" was significantly different in two gender groups, with CR $= -2.02$ (the absolute CR value $> 1.96$, $p < 0.05$). Specifically, the effect of BPNS on negative affect (shown in Fig. 2) was greater in female adolescents ($\beta = -0.60$, $t = -13.55$, $p < 0.001$) than that in male adolescents ($\beta = -0.51$, $t = -9.35$, $p < 0.001$). The same method was applied to test whether the other path coefficients significantly differ between male and female adolescents, and it revealed no significant gender differences.

A bias-corrected bootstrap approach (a bootstrap sample of 2,000) was used to test the mediating effect in male and female groups. Results showed that the mediating role of negative affect was significant in the male group (mediating effect index $= -0.30$), with the 95% CI ($-0.38$, $-0.13$) not containing zero. Meanwhile, the mediating effect of negative affect was also significant in the female group, with a 95% CI of ($-0.51$, $-0.28$), and this indirect effect (mediating effect index $= -0.40$) was greater. These results confirmed the

mediating effect of negative affect in both male and female adolescents, but the mediating effect is greater in female adolescents.

## DISCUSSION

This study aimed to explore the effect of BPNS on aggressive behavior, as well as the mediating role of affect and the moderating role of gender in Chinese adolescents. Structural equation modeling results showed that BPNS was negatively associated with aggressive behavior, which is consistent with previous research (*Choe & Read, 2019*). These findings indicate that BPNS is an important protective factor of adolescents' aggressive behavior. Therefore, SDT can be used as a theoretical framework to prevent and reduce adolescent aggressive behavior.

### The mediating role of negative affect

Our results showed that negative affect mediated the relationship between BPNS and aggressive behavior, which supports GAM (*Anderson & Bushman, 2002*), and the cognitive-neoassociationistic model (*Berkowitz, 1993*). To the best of our knowledge, the full paths of the relationships among BPNS, affect, and aggressive behavior in our mediating model have not been investigated before, especially among Chinese adolescents. However, several paths have been examined separately in previous literature. Specifically, our findings are consistent with previous research (*Schutte & Malouff, 2021*) indicating that BPNS is a significant and negative predictor of negative affect. Moreover, our results showed that negative affect positively predicted aggressive behavior in adolescents. According to previous studies, affect is an important factor influencing aggressive behavior (*Lazarus, 2000*; *Laible, Murphy & Augustine, 2014*). High levels of negative affect make individuals more susceptible to a range of emotional biases, which would further lead to aggressive behavior (*Burt & Donnellan, 2008*). Thus, emotion plays an important role in psychological perception and behavior (*Yang, Li & Liu, 2021*). Consequently, adolescents with low basic psychological needs satisfaction may arise higher negative affect, which, in turn, leads to more aggressive behaviors.

Besides negative affect, our study also included positive affect in the model for additional analysis. The results showed that BPNS was significantly associated with positive affect, but the relationship between positive affect and aggressive behavior was not significant. These findings indicates that BPNS plays an indirect role in adolescent aggression primarily through negative affect rather than positive affect. Our findings further provide empirical support for the bivariate model of positive and negative affect (*Larsen, McGraw & Cacioppo, 2001*), which states that positive affect and negative affect are separable constructs. Meanwhile, compared to the inhibitory effect of positive affect on aggressive behavior, negative affect has more detrimental effect on aggressive behavior and make adolescents showing more maladaptive behaviors (*i.e.,* aggressive behavior). Thus, to effectively reduce adolescents' aggressive behavior, it is better to take measures to cope with individuals' negative affect rather than increasing their positive affect.

## Gender difference in mediation effect

Our study also found that the mediating role of negative affect between the relationship of BPNS and aggressive behavior in Chinese adolescents is gender-specific. Multi-group comparison analysis indicated that the effect of BPNS on negative affect was greater in female group and, in turn, the indirect effect of BPNS on aggressive behavior through negative affect was stronger for females. This findings confirmed our hypothesis and are consistent with previous studies conducted among college students which provided that females were more susceptible to negative affect (*Zhao et al., 2020*). With respect to basic psychological needs (*Ryan & Deci, 2008*), failure to have these needs satisfied has a particularly pronounced effect in female adolescents, partly as a result of the emotional instability that comes with adolescence, and also because females tend to be more emotionally susceptible. The upshot of this is that adolescent females are more vulnerable to the impact of negative affect which may lead to displays of aggressive behavior.

In contrast, our study did not find any gender differences on the path "negative affect → aggressive behavior", and the path "BPNS → aggressive behavior", even though females are more subject to negative affect. In our study, YSR was used to measure the general level of aggression in adolescents without subdividing aggression into different types of aggressive behaviors. Future research may focus on specific aggression to provide further insights into the gender differences of aggression in adolescents.

## Limitations and implications

This study has several limitations which are now discussed. First, only self-reported measures were used to assess all our variables. Adopting multiple methods to replicate the present results is necessary. Second, the present study is only correlational, which prevents us from providing a strong causal inference. Future studies should adopt a longitudinal design to enhance the causality of our findings. Third, participants in our study were selected from junior middle school in the southeast of China. Therefore, whether our finding could generalize to other sample groups from other cultures still need further research. Finally, prior research has found that specific types of BPNS may have different influences on individuals' behaviors (*e.g.*, *Granero-Gallegos et al., 2019*), whereas the current study mainly focused on the overall effect of BPNS on aggressive behavior. Thus, future research would benefit from further examining whether three different BPNS are associated with negative affect and aggressive behaviors in different patterns.

Despite these limitations, this study makes some contributions and provides several implications for the preventions and interventions of aggressive behaviors in adolescence for the future research. First, to our knowledge, it is the first study to elaborate the potential mechanism that links BPNS with aggressive behavior in Chinese adolescents. Our findings offered the first step for intervention efforts that target adolescent aggression. By demonstrating that BPNS is a significant protective factor against negative affect and aggressive behavior, our study suggests that satisfying basic psychological needs of adolescents may be one of the fundamental ways to prevent them from aggressive behavior. Thus, providing a supportive living environment to improve adolescents' need satisfaction and enhance their sense of autonomy, competence, and relatedness could serve as a

prevention of aggression problems. Second, given the mediating role of negative affect and gender differences, it is important for educators and parents to help adolescents coped with negative affect appropriately, particularly for girls. As high levels of negative affect may be identified as a risk factor of aggressive behavior (*e.g.*, *Gutierrez-Cobo et al., 2018*), thus dealing with negative emotions properly could be a useful way for the prevention of aggressive behavior in adolescence. Third, basic psychological need satisfaction (BPNS) and basic psychological need frustration are considered as distinct but relatable constructs (*Vansteenkiste & Ryan, 2013*), which can substantially account for both the "dark" and "bright" side of people's functioning (*Ryan & Deci, 2000*). In addition, recent research has shown that need frustration is associated with problem behaviors and various psychological adjustment problems such as depression and negative affect (*e.g.*, *Brenning et al., 2021*). Thus, the additional assessment of need frustration may allow us to better account for various maladaptive behaviors (*e.g.*, aggressive behavior) in future research.

Last but not least, latent-variable-based SEM and item parceling technique were applied in our data analyses. Parceling is a technique with great potential but accompanied by a number of controversies. On the one hand, parceling can provide psychometric and modeling-related benefits. For instance, item parceling can stabilize parameter estimates and improve the model fit (*e.g.*, *Matsunaga, 2008*). Meanwhile, parceling often helps mitigate the problem of nonnormality (*Hau & Marsh, 2004*). On the other hand, the use of parceling may have the issues of potential induction of estimation bias under certain circumstances (*Matsunaga, 2008*). But some researchers argued that parceling methods were indistinguishable from, or slightly better than, the item-based method in terms of estimation bias, and even the all-item-parcel approach's bias was "fairly minor" (*e.g.*, *Bandalos & Finney, 2001*). Even though the estimate bias is not a concern in current study, caution is suggested to use parceling approach in future research.

## CONCLUSIONS

This study explored the relationship between BPNS and aggressive behavior in Chinese adolescents. Our results showed that BPNS was significantly and negatively associated with aggressive behavior in adolescents. Negative affect played a mediating role in the relationship between BPNS and aggressive behavior, and gender played a moderating role in the mediating model. Specifically, the prediction from BPNS to negative affect was greater in females than in males and, equally, the mediating effect of negative affect was greater in females. Our findings provide theoretical and empirical evidence for psychological interventions to reduce aggression in adolescents.

### Funding

This work was supported by the Teaching Reform Research and Practice Project of Henan University (No. YB-JFZX-2022-06). The funders had no role in study design, data collection and analysis, decision to publish, or preparation of the manuscript.

## Grant Disclosures

The following grant information was disclosed by the authors:

Teaching Reform Research and Practice Project of Henan University: No. YB-JFZX-2022-06.

## Competing Interests

The authors declare there are no competing interests.

## Author Contributions

- Fen Dou conceived and designed the experiments, performed the experiments, analyzed the data, prepared figures and/or tables, authored or reviewed drafts of the article, and approved the final draft.
- Qinglin Wang conceived and designed the experiments, performed the experiments, analyzed the data, prepared figures and/or tables, authored or reviewed drafts of the article, and approved the final draft.
- Minghui Wang conceived and designed the experiments, performed the experiments, analyzed the data, prepared figures and/or tables, authored or reviewed drafts of the article, and approved the final draft.
- Entao Zhang conceived and designed the experiments, performed the experiments, analyzed the data, prepared figures and/or tables, authored or reviewed drafts of the article, and approved the final draft.
- Guoxiang Zhao conceived and designed the experiments, performed the experiments, analyzed the data, prepared figures and/or tables, authored or reviewed drafts of the article, and approved the final draft.

## Human Ethics

The following information was supplied relating to ethical approvals (i.e., approving body and any reference numbers):

The Psychology Research and Ethics Committee at Henan University in China to carry out the study within its facilities.

## Ethics

The following information was supplied relating to ethical approvals (i.e., approving body and any reference numbers):

The Psychology Research and Ethics Committee at Henan University in China to carry out the study within its facilities.

## Data Availability

The raw data is available in the Supplemental File.

## Supplemental Information

Supplemental information for this article can be found online at http://dx.doi.org/10.7717/peerj.16372#supplemental-information.

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
