# Peer review of "Basic psychological need satisfaction and aggressive behavior: the role of negative affect and its gender difference"

_PeerJ, doi:10.7717/peerj.16372_

## Round 0.1 · original submission · Minor Revisions

Two reviewers give substantial comments and suggestions. You need to read their reviews and revise them accordingly carefully. Before resubmission, please check every question/comment with corresponding changes in your manuscript and mark them with color.

Reviewer 1 ·

Basic reporting

Many thanks for reviewing this manuscript on "Basic psychological need satisfaction and aggressive behavior: The role of negative affect and its gender difference." The authors presented a test of the relationship between basic psychological need satisfaction and aggressive behavior via the mediating role of negative affect and the moderating role of gender, using a sample of junior students. The methods and findings are adequate; however, there are several critical issues that prevent their work from being promoted, and these need to be further clarified and addressed. These are outlined below.

1. A lack of solid reasoning for supporting the topic and proposed model: The authors have provided some good reasons for the importance of aggressive behavior. However, the reasoning for why they should focus on BPNS, PANA, and gender differences is too weak and insufficient. There are many factors that have been proven to influence aggressive behavior, such as personality, brain mechanisms, or even genetics. The authors need to provide more convincing arguments to support their proposed model. Additionally, they should justify why the mediating role of PANA and moderating role of gender are specifically important mechanisms in the context of existing theories and literature and should be addressed in this study. A well-reasoned approach might help to identify the research gap and clarify the value and contributions of their work.
2. Insufficient review and discussion of relevant literature: While the authors briefly introduced the importance of aggressive behavior and BPNS in their study, they seem to be unaware of much of the relevant literature and theories related to both aggressive behavior and BPNS in the field of psychology. For example, there is no information about the various types of aggression in the introduction or literature review sections, even though they mention a focus on specific aggression in future research. The hypotheses and arguments are primarily based on reporting a few past results without specific theoretical justifications. It is suggested that the authors draw on existing literature and theories in psychology to provide a more comprehensive and complete theoretical foundation for the relationships among their focal variables.
3. Lack of a main perspective (or theory) to build the proposed model: The authors have used too many different theories, including SDT, COR, cognitive-neoassociationistic model, and role theory, to develop their hypotheses. However, the arguments are relatively fragmented, logic-jumping, and lacking a comprehensive organization. As the focus of this manuscript is on testing mediation and moderation effects, it is advisable to draw on one or combine two theories that can adequately cover their focal variables and further develop their hypotheses. Theories relevant to self-control, self-regulation, or COR might be appropriate to guide their proposed model, as they can explain aggressive behavior when individuals are low in psychological resources that help maintain self-control behaviors.
Reference:
Buker, H. (2011). Formation of self-control: Gottfredson and Hirschi's general theory of crime and beyond. Aggression and Violent Behavior, 16(3), 265-276.

4. A lack of solid reasoning for the chosen subject: The authors have not provided any reasons for why junior students, particularly from China, were chosen as the important target for investigating aggressive behaviors. The current reasoning mainly explains the background and current situation regarding focal variables. It is advisable to provide more concise reasons to justify why junior students are an important target for studying aggressive behaviors, considering that the occurrence of aggressive behaviors might vary in different subjects.

Experimental design

5. Methodological concerns: While the authors have used cross-sectional data in this study and conducted good data analysis, one major concern is the item parceling in their SEM analysis. The manuscript lacks clarity and explanation regarding the theory and principles behind the item parceling approach. The authors should provide more details to explain the uses and procedures of item parceling and discuss its implications for construct validity after parceling the variables (e.g., BPNS: competence, autonomy, relatedness).

Validity of the findings

6. Discussions are limited around the current results: The manuscript would benefit from more consideration, conversation, and discussion of the theoretical contribution, or at least broadening the discussion section, with a focus on psychology literature.

Additional comments

Minor Comments:
(1)L.90-94: The contents do not explain the moderating effect but the direct effect of gender. It is advisable to reconsider and modify this paragraph.
(2)L.95-101: The arguments in this section contradict L.82-87. While L.82-87 suggest a stronger effect for girls, L.95-101 indicates that the authors cannot make clear and specific assumptions about the gender effect on adolescents' aggressive behavior. This inconsistency should be addressed and clarified.
(3)It is advisable to list all the hypotheses in the manuscript.
(4)L.253-259: This paragraph does not provide any new insights for readers and merely repeats old information. Additionally, the sentence "the mediating effect of positive affect is not significant, mainly because positive affect has no significant effect" is redundant and does not offer any further explanations for readers. It is advisable to reconsider and modify this paragraph to enhance its contribution to the manuscript.
(5) L.281-283: For this point “the present study is only correlational, which prevents us from providing 
a strong causal inference. Experimental or longitudinal study is needed to enhance the causality of 
our findings.“, I am not sure about how the researcher can conduct an experimental study to manipulate BPNS.

·

Basic reporting

The study aimed to discover the relationships between basic psychological need satisfaction and aggressive behaviors as well as the mediating roles of negative affect and genders. The manuscript is well-structured, the rationale and hypothesis for the study were theoretically and empirically sound. The methods and analyses were adequate. In general, this is considered a solid research. Nonetheless, there are several issues that should be addressed.

Experimental design

It was mentioned that previous studies rendered inconsistent results regarding the direct and indirect aggressive behaviors between males and females in the introduction section so it would be contributive to discover the sex differences of aggressive behaviors. It is disappointing to find that no discussion was made regarding what types of aggressive behaviors were more likely to be involved by males or females which resulted from the choice of measurement. It is puzzling why chose such a measurement since it was unable to solve or shed light on the inconsistency in previous findings.

Previous studies have found a direct link between BPNS and aggressive behaviors but not in this study. An explanation of such inconsistency is expected but not provided in the manuscript.

Validity of the findings

The suggestion of emotional management to ease aggressive behaviors may be overreaching based on the findings of the study.

It has been found that the satisfaction of autonomy, competence, and relatedness need may have different influences on/correlation with motivation types and adaptive/maladaptive behaviors. It would be interesting to separate three different BPNs to see how they interact with different negative affect and aggressive behaviors individually.

Additional comments

1. Line 22, “effect” should be “affect.”
2. Please specify the factors and item numbers of each measurement and list example items as well.
3. Line 241 “in indicating” should be “indicating.”
4. Figure 1 is not clear enough for reading.
5. Please provide the validity and reliability of original measurements in the methods section.
6. The dark path of SDT has been extensively discussed in recent studies and has concluded that the unsatisfaction of BPNs and the frustration of BPNs rendered different influences on motivation tendencies and adaptive/maladaptive behaviors, i.e., the frustration of BPNs have stronger predictability on/correlation with maladaptive behaviors than unsatisfaction of BPNs. While only the satisfaction of BPNs was measured and examined in the study, it is recommended to mention such limitations and make suggestions for future studies accordingly.

---

## Round 0.2 · Minor Revisions

The manuscript has been revised to an acceptable feature. However, the first reviewer has two questions. Please refer to it and revise it accordingly. I will make a final decision after your reply. Also, I have seen there are many places with editing problems, such as grammatical errors, spacing, and APA formats. Please recheck it before submission.

**Language Note:** The Academic Editor has identified that the English language must be improved. PeerJ can provide language editing services - please contact us at copyediting@peerj.com for pricing (be sure to provide your manuscript number and title). Alternatively, you should make your own arrangements to improve the language quality and provide details in your response letter. – PeerJ Staff

Reviewer 1 ·

Basic reporting

The revised manuscript has significantly improved, and here I list two comments/questions to the authors, requiring further clarification and justification.

1. While the authors argued that “BPNS is a situational factor”, I am not sure about such a perspective. In general psychology, BPNS has been regarded as an individual psychological outcome. As the definition of BPNS is vital in explaining their hypothesized model, it is advisable to clarify the definition of BPNS.
2. For the hypothesis, the authors argued that “due to a lack of theory and empirical research about positive affect, we formulated no specific hypotheses about positive affect”. However, the COR theory actually offers sufficient support for explaining the mechanism of BPNS-PA-Aggressive behavior. I would therefore suggest they further build this psychological path based on COR, and list a specific hypothesis.

Experimental design

None

Validity of the findings

None

Additional comments

None

·

Basic reporting

Thanks for the opportunity to review this manuscript. The authors appear to have substantially addressed my concerns. Please give the entire manuscript a careful proofread/edit to eliminate typos, writing and formatting issues etc. Besides, please ensure all arguments are supported with appropriate references.

Experimental design

no comment

Validity of the findings

no comment

---

## Round 0.3 · accepted · Accept

Your manuscript has been revised appropriately, following the reviewers' suggestions. However, I found there are many grammatical and editing errors. I attached a sample, which I revised in the front line 44. Please refer to it and check it again.